# Reference Standards for Digital Infrared Thermography Measuring Surface Temperature of the Upper Limbs

**DOI:** 10.3390/bioengineering10060671

**Published:** 2023-06-01

**Authors:** Seong Son, Byung Rhae Yoo, Ho Yeol Zhang

**Affiliations:** 1Department of Neurosurgery, Gil Medical Center, Gachon University College of Medicine, Incheon 21565, Republic of Korea; sonseong44@gmail.com; 2Department of Neurosurgery, National Health Insurance Service Ilsan Hospital, Yonsei University College of Medicine, Ilsan 10444, Republic of Korea

**Keywords:** infrared rays, reference standard, skin temperature, thermography, upper limb

## Abstract

(1) Background: although digital infrared thermographic imaging (DITI) is used for diverse medical conditions of the upper limbs, no reference standards have been established. This study aims to establish reference standards by analyzing DITI results of the upper limbs. (2) Methods: we recruited 905 healthy Korean adults and conducted thermography on six regions (dorsal arm, ventral arm, lateral arm, medial arm, dorsal hand, and ventral hand region). We analyzed the data based on the proximity of regions of interest (ROIs), sex, and age. (3) Results: the average temperature (°C) and temperature discrepancy between the right and the left sides (ΔT) of each ROI varied significantly (*p* < 0.001), ranging from 28.45 ± 5.71 to 29.74 ± 5.14 and from 0.01 ± 0.49 to 0.15 ± 0.62, respectively. The temperature decreased towards the distal ROIs compared to proximal ROIs. The average temperatures of the same ROIs were significantly higher for men than women in all regions (*p* < 0.001). Across all regions, except the dorsal hand region, average temperatures tended to increase with age, particularly in individuals in their 30s and older (*p* < 0.001). (4) Conclusions: these data could be used as DITI reference standards to identify skin temperature abnormalities of the upper limbs. However, it is important to consider various confounding factors, and further research is required to validate the accuracy of our results under pathological conditions.

## 1. Introduction

Digital infrared thermographic imaging (DITI) has been utilized as an ancillary diagnostic method for various medical conditions related to the upper extremities. For instance, cervical radiculopathy; peripheral nerve entrapment syndrome (e.g., carpal tunnel syndrome); rheumatic disease (e.g., Raynaud’s disease); complex regional pain syndrome; tendinopathy; hand arthritis (e.g., psoriatic arthritis); and skin cancer can be confirmed by DITI [1,2,3,4,5,6,7,8,9,10,11,12,13]. The hypo-radiant (hypothermia) or hyper-radiant (hyperthermia) regions can be identified by comparing the temperature between the right and left arms or with empirical normal ranges. However, the significant temperature differences between both sides and deviations from normal ranges have not been established. Furthermore, body surface temperatures are influenced by the measurement environment, ana-tomical area, and subject characteristics, which presents a challenge in deriving reference values from the existing literature [14,15].

Consequently, previous attempts to establish reference standards for DITI using systematic reviews based on meta-analysis or machine learning methods have been limited in providing detailed standard DITI values [10,16,17,18]. Although a recent study suggested a correct differential diagnosis process for Raynaud’s phenomenon in the hand using a deep convolutional neural network, a type of deep learning method, its small data set had limitations [19]. To address this gap and establish scientific reference standards for DITI, we investigated the DITI results of several body regions, including the lower limbs, upper limbs, trunk, and face, in a large cohort of subjects under a controlled protocol. We previously reported DITI reference standards for the lower limbs [16]. In the present study, we provide DITI reference standards for the upper limbs, including the hands.

## 2. Materials and Methods

### 2.1. Study Design, Ethics, and Sample Size

This multi-center, single-arm, open label study was conducted in compliance with global/local ethics. The Institutional Review Board of three hospitals approved the study protocol. In addition, this clinical trial was registered with the Clinical Research Information Service of Korea (CRIS, http://cris.nih.go.kr, accessed on 1 April 2023, number KCT0006880) [20].

The sample size for this study was determined to be 922 participants, with a dropout rate of 15%, based on the equation provided in our previous report [20]. The equation used was as follows:$$n\;=\;\frac{{\theta \left(1\;-\;\theta \right)}^{2}z_{\alpha /2}^{2}}{d^{2}}$$

(Population proportion [θ] = 0.85; margin of error [d] = 0.025; and significance level of 5% with confidence level of 95%).

### 2.2. Subjects Recruitment

Healthy Korean adults were recruited through open announcements at three hospitals between March 2018 and December 2020.

The general qualifications for participation in the DITI examination of whole body, including the upper extremities, lower extremities, trunk, and face, were as follows: (1) age between 20 and 69; (2) the absence of any potential contraindication for DITI (e.g., pregnancy or claustrophobia); (3) the ability to maintain posture during the test; and (4) the absence of any other issue considered by the investigator to adversely influence DITI results [20].

To minimize potential confounders, the following criteria were adopted: (1) no specific medical history, such as cervical spine disease, diabetes mellitus, peripheral neuropathy or entrapment, joint disorder of the arm or hand, previous history of cervical spine or upper extremity surgery; and (2) currently no pain or skin abnormality of the upper extremity. 

Initially, 922 healthy Korean adults were enrolled. However, 17 patients were excluded due to test failure or withdrawal of consent. Therefore, the results from 905 participants were included in the analysis [20]. All testing was performed under controlled conditions after obtaining informed consent.

### 2.3. Equipment and Examination Protocol

DITI testing was performed using the same equipment and methodology as we previously described [20]. The Iris-XP Digital infrared imaging system (Medicore, Seoul, Republic of Korea) was employed to obtain the images. Prior to the test, subjects were required to spend 20 min in the test room to acclimatize to the controlled room conditions, which maintained a temperature of 20.0–23.0 °C and humidity of 30–75%. The distance between the subject and the equipment was set at 1.5 m.

During the DITI imaging, upper limbs were divided into six sections based on empirical classification, which facilitated classification into various two-dimensional planes, including the front, back, and lateral sides. The upper limbs were tested in a neutral standing position, encompassing six regions, including the dorsal, ventral, lateral, and medial arm regions, as well as the dorsal and ventral hand regions. Regions of interest (ROIs) were manually marked on both sides and comprised 18 ROIs in the dorsal arm region, 19 ROIs in the ventral arm region, 19 ROIs in the lateral arm region, 7 ROIs in the medial arm region, 26 ROIs in the dorsal hand region, and 26 ROIs in the ventral hand region (Figure 1). Five certified examiners performed the testing using a diagram that provided details of measurements and ROI locations.

### 2.4. Statistical Analysis

SPSS version 27.0 (IBM Corporation, Armonk, NY, USA) was used for statistical analysis. Data were checked for normality using the Kolmogorov–Smirnov test, and the results were expressed as mean ± standard deviation (SD), mean with 95% confidence intervals (CI), or median with range. One-way analysis of variance (ANOVA), paired *t*-test, or linear regression analysis were applied according to the purpose of analysis. A *p* value of less than 0.05 was considered statistically significant.

## 3. Results

### 3.1. Basic Demographic Data

The average age of all subjects (*n* = 905) was 42.86 ± 12.87 years, and the male-to-female ratio was 411:405. The sample consisted of 183 individuals (97 males and 86 females) in their 20s, 213 (108 males and 105 females) in their 30s; 228 (109 males and 119 females) in their 40s; 177 (65 males and 112 females) in their 50s; and 104 individuals (32 males and 72 females) in their 60s [20]. 

### 3.2. Average Temperatures and Temperature Discrepancies between Right and Left Sides for Each ROI (°C) 

The average temperatures and average temperature discrepancies between the right and left (ΔT, temperature of right side−temperature of the left side) of each ROI were as follows: 29.22 ± 5.71 (range, 28.56 ± 5.42–30.24 ± 5.90) and 0.06 ± 0.49 (range, −0.08 ± 0.64–0.25 ± 0.48) in the dorsal arm region; 29.63 ± 5.74 (range, 29.08 ± 5.69–30.51 ± 6.00) and 0.01 ± 0.49 (range, −0.18 ± 0.49–0.18 ± 0.43) in the ventral arm region; 28.45 ± 5.71 (range, 27.59 ± 5.39–29.53 ± 5.90) and 0.15 ± 0. 62 (range,–0.20 ± 0.76–0.52 ± 0.78) in the lateral arm region; 28.84 ± 5.57 (range, 28.00 ± 5.23–29.46 ± 5.79) and 0.04 ± 0.48 (range, −0.10 ± 0.69–0.11 ± 0.42) in the medial arm region; 29.74 ± 5.14 (range, 28.08 ± 4.89–31.27 ± 5.53) and 0.06 ± 0.59 (range, −0.17 ± 0.57–0.26 ± 0.49) in the dorsal hand region; and 28.62 ± 5.44 (range, 27.08 ± 5.14–29.98 ± 5.60) and 0.07 ± 0.69 (range, −0.15 ± 0.57–0.20 ± 0.88) in the ventral hand region (Table 1, Table 2, Table 3, Table 4, Table 5 and Table 6).

The average temperatures of the ROIs differed significantly between the six regions (*p* < 0.001, ANOVA) (Figure 2). 

The overall median of the average of the absolute value of the ΔT (|ΔT|, $\sqrt{{\left[\mathrm{temperature}\;\mathrm{of}\;\mathrm{the}\;\mathrm{right}\;\mathrm{side}-\mathrm{left}\;\mathrm{side}\right]}^{2}}$) was 0.10 (range, 0.00–0.52). Additionally, |ΔT| values differed significantly between the six regions (*p* < 0.001, ANOVA). In particular, the |ΔT| in the lateral arm region was significantly larger (median 0.14 [range, 0.05–0.52]) than that of any other regions (Figure 3).

### 3.3. Subgroup Analysis Based on ROI Location

Distally located ROIs had significantly lower average surface temperatures, except for the lateral arm region, as indicated by the regression analysis. The relationship between the average temperature of each ROI and the proximity of ROIs were as follows: dorsal arm region temperature = 29.916 − (0.181 × ROIs) (*p* < 0.001, R^2^ = 0.480); ventral arm region temperature = 29.915 − (0.070 × ROIs) (*p* = 0.012, R^2^ = 0.162); lateral arm region temperature = 28.684 − (0.063 × ROIs) (*p* = 0.132, R^2^ = 0.062); medial arm region temperature = 29.591 − (0.188 × ROIs) (*p* < 0.001, R^2^ = 0.750); dorsal hand region temperature = 31.474 − (0.399 × ROIs) (*p* < 0.001, R^2^ = 0.799); and ventral hand region temperature = 29.855 − (0.284 × ROIs) (*p* < 0.001, R^2^ = 0.429) (Figure 4).

The average |ΔT| values of the ROIs did not show any specific trend. The relationship between the average |ΔT| of each ROI and the proximity of ROIs were as follows: dorsal arm region |ΔT| = 0.074 + (0.006 × ROIs) (*p* = 0.535, R^2^ = 0.024); ventral arm region |ΔT| = 0.031 + (0.013 × ROIs) (*p* = 0.090, R^2^ = 0.160); lateral arm region |ΔT| = 0.299 − (0.030 × ROIs) (*p* = 0.066, R^2^ = 0.185); medial arm region |ΔT| = 0.090 − (0.001 × ROIs) (*p* = 0.855, R^2^ = 0.007); dorsal hand region |ΔT| = 0.157 − (0.012× ROIs) (*p* = 0.117, R^2^ = 0.099); and ventral hand region |ΔT| = 0.078 + (0.008 × ROIs) (*p* = 0.225, R^2^ = 0.061) (Figure 5).

### 3.4. Subgroup Analysis Based on Sex

The average temperatures of each same ROI were significantly higher in male subjects compared to female subjects across all regions and age groups (*p* < 0.001, paired *t*-test). The average temperatures for each region were as follows: 30.31 ± 5.43 (range, 29.45 ± 5.83–32.51 ± 4.59) for men and 28.32 ± 5.94 (range, 26.06 ± 5.85–29.59 ± 5.82) for women in the dorsal arm region; 30.72 ± 5.47 (range, 29.9 ± 5.93–32.89 ± 4.55) for men and 28.73 ± 5.97 (range, 26.51 ± 5.92–30.05 ± 5.86) for women in the ventral arm region; 29.66 ± 5.45 (range, 28.76 ± 5.89–31.67 ± 4.60) for men and 27.44 ± 5.92 (range, 25.13 ± 5.85–28.81 ± 5.73) for women in the lateral arm region; 29.84 ± 5.33 (range, 29.11 ± 5.72–31.84 ± 4.56) for men and 28.00 ± 5.78 (range, 25.80 ± 5.64–29.34 ± 5.72) for women in the medial arm region; 30.51 ± 4.93 (range, 29.67 ± 5.33–32.38 ± 3.83) for men and 28.81 ± 5.38 (range, 27.34 ± 5.21–29.38 ± 5.25) for women in the dorsal hand region; and 29.71 ± 5.22 (range, 29.12 ± 5.14–31.30 ± 4.55) for men and 27.72 ± 5.63 (range, 25.48 ± 5.43–29.14 ± 5.67) for women in the ventral hand region (Table 7). 

The temperature difference between both sexes ranged from 0 °C 63 (95% CI, 0.24–1.01) to 3.80 °C (95% CI, 3.49–4.10), depending on region and age group (Figure 6).

### 3.5. Subgroup Analysis Based on Age Group

The average temperatures of the ROIs significantly increased with age, particularly in individuals in their 30s and older in all regions, except for the dorsal hand region (*p* < 0.001, ANOVA). However, subjects in their 20s had a higher surface temperature than those in their 30s (*p* < 0.001, paired *t*-test). Consequently, surface temperatures were lowest for those in their 30s for all regions, except for the dorsal hand region (*p*< 0.001, ANOVA post hoc analysis) (Table 7 and Figure 7).

## 4. Discussion

### 4.1. Average Temperature and |ΔT| of Each ROI

The average surface temperature of the upper limbs varied depending on regions and ROIs, ranging from 27.08 ± 5.14 °C to 31.27 ± 5.53 °C, which is consistent with the results found in our previous study on the lower limbs. However, the average temperatures of the upper limbs were higher than those of the lower limbs (range, 24.60 ± 5.06 °C–27.75 ± 5.76 °C) [20]. We believe this difference is due to the different distance from the heart [21,22,23].

The average of |ΔT| of each ROI also varied by regions and ROIs, ranging from 0.00 to 0.52 °C. However, these values are smaller compared to the reference standards of the lower limbs, where |ΔT| reached 0.76 °C [20]. Nevertheless, the practical value of |ΔT| of the upper limbs was not within 0.1–0.3 °C, which is considered the normal range, based on the previous consensus [15,24,25].

These data for each ROI provide reference standards for DITI of the upper limbs. Clinically significant cold/hot areas, or specific areas with a significant difference between both sides, can be detected based on comparative analysis between the practical patient’s image and these data. Additionally, the detailed ROIs in this study can be modified simply during the processing of DITI capture in actual clinical practice. However, accurate diagnosis requires a comparison of each ROI, not just the averages of whole regions, as normal ranges of surface temperature and ΔT values vary depending on region or ROI.

Various intrinsic factors, such as sex, age, fat percentage, and menstrual cycle stage, and extrinsic factors, such as test environment, testing time, and season, can influence the results [26,27,28,29]. As a result, actual DITI measurements may fall outside the reference standard values. Therefore, further clinical studies are necessary to validate these data by comparing the DITI results of patients with specific diseases with those of healthy subjects.

### 4.2. Subgroup Analysis Based on Proximity of ROIs, Sex, and Age Group

A significant correlation was found between the proximity of ROIs to the body core and surface temperature in all regions, except for the lateral arm region, i.e., surface temperatures decreased from the proximal regions to the distal ends. These findings are consistent with a previous suggestion that the surface temperatures of peripheral regions (e.g., hand or foot) tend to be cooler than central regions (e.g., trunk or proximal arm), due to the greater distances from the main thermal organs, such as the heart, large vessels, or viscera [20,21,22,23]. This tendency was found to be greater for the upper limbs than lower limbs, which can be attributed to the fact that the upper limbs are closer to the heart than the lower limbs, resulting in the proximal region being distinctly warmer than the distal region [20].

However, the |ΔT| values showed no specific trend based on proximity of ROIs to the body core, which is different from a previous suggestion that the |ΔT| values are higher in the distal regions compared to the proximal regions [2,15]. This was also found in the lower limbs in a previous study [20].

Surface temperatures were found to be higher for men in all regions, which is consistent with suggestions that surface temperatures are lower for women due to the insulating effect of thicker subcutaneous fat [30]. This trend was also observed for the lower limbs [20].

Surface temperatures increased significantly with age from the 30s, except for the dorsal hand region, which is also consistent with previous research on the lower extremities [20]. These trends may be caused by age-related diminished vasoconstriction by the sympathetic nervous system [31,32,33]. The paradoxical higher temperatures for those in their 20s can be explained by a higher basal metabolic rate and less subcutaneous fat [34,35].

### 4.3. Limitations and Significance

Establishing reference standards for DITI is a complex task due to the influence of various confounding factors on skin temperature and DITI measurements [36]. Therefore, it is important to acknowledge several limitations of our study. First, body fat percentage is a significant confounding factor in measuring body surface temperature using DITI [37] However, we did not collect data on body fat percentage or calculate the body mass index based on body weight and height. Secondly, daily biorhythms, such as menstrual cycle status, menopause, sleep patterns, and emotional stress, can also impact DITI results [38,39]. Unfortunately, we did not account for these factors during the examination. Thirdly, various extrinsic factors, including diurnal testing time, season, and ethnicity were not considered [39]. Moreover, even though we recruited healthy adult volunteers based on questionnaire responses, the process does not guarantee the exclusion of specific diseases or conditions that may have affected DITI results.

Nonetheless, the study holds value due to its large sample size and the use of a consistent protocol for measurements. It represents the first attempt to establish reference standards for DITI measurements of the upper limbs.

## 5. Conclusions

The findings of this study provide a basis for establishing reference standards for DITI measurements of surface temperatures in the upper extremities. These standards can aid physicians in making objective diagnoses by comparing patient DITI results with the provided data. However, it is crucial to consider the influence of various confounding factors in surface temperature measurements. Moreover, before confirming the results, it is important to consider several parameters, including the specific location of the ROIs, sex, and age.

## Figures and Tables

**Figure 1 bioengineering-10-00671-f001:**
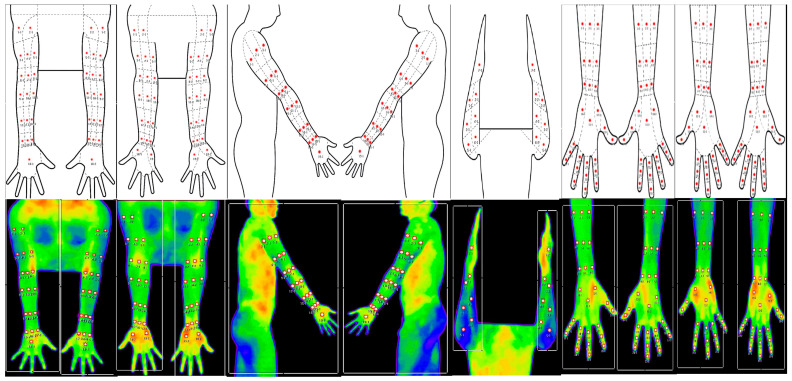
Schematics and actual photographs showing the regions of interest in the dorsal arm, ventral arm, lateral arm, medial arm, dorsal hand, and ventral hand of the upper limbs. The red dots represent each region of interest, and the related numbers are the serial numbers assigned to each regions of interest.

**Figure 2 bioengineering-10-00671-f002:**
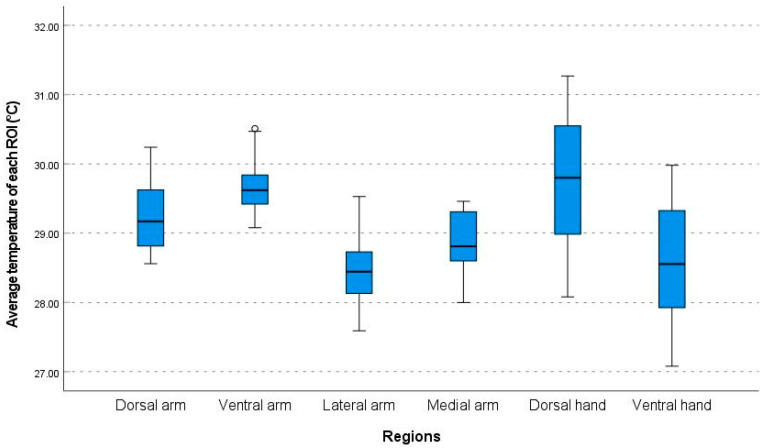
Comparison of the average temperatures of each ROI in the six regions. ROI: region of interest; and ⸰ indicates a label out of range.

**Figure 3 bioengineering-10-00671-f003:**
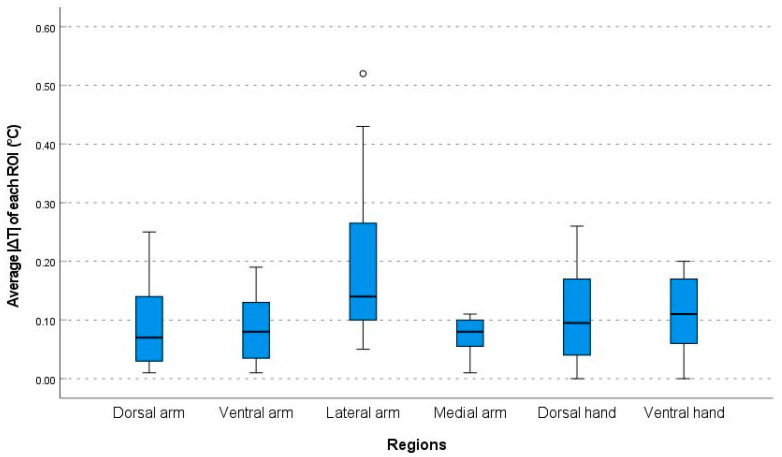
Comparison of the absolute value of the temperature discrepancies between the right and left side (|ΔT|) of each ROI in the six regions. ROI: region of interests; and ⸰ indicates a label out of range.

**Figure 4 bioengineering-10-00671-f004:**
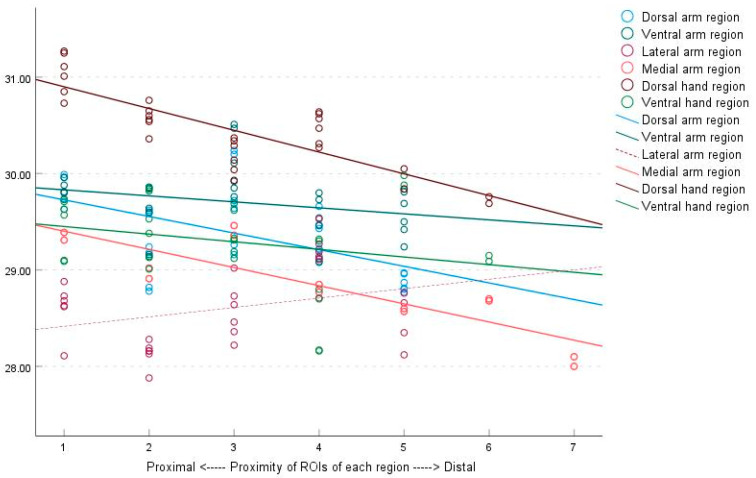
Correlation between the average temperatures of each ROI and the proximity of ROIs to the body core of the six regions, as determined by linear regression analysis. The trend observed in the figure demonstrates a gradual decrease in surface temperature from proximal to distal for each ROI, except for the lateral arm region. ROIs, regions of interest.

**Figure 5 bioengineering-10-00671-f005:**
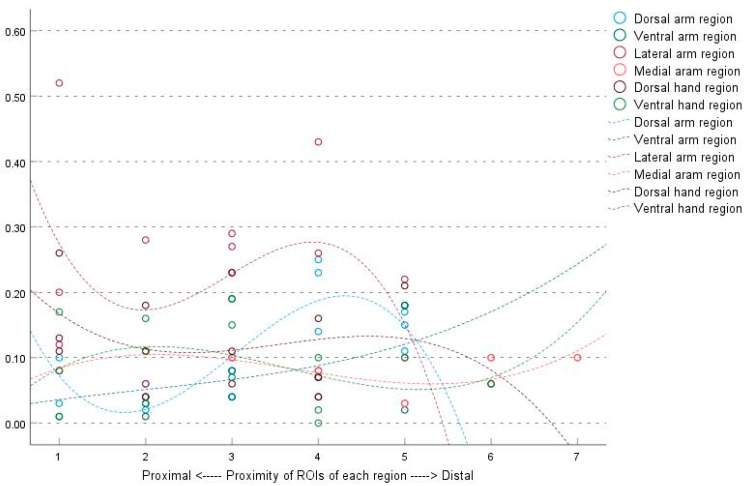
Correlation between right to left temperature discrepancies and the proximity of ROIs to the body core of the six regions, as determined by linear regression analysis. No perceptible correlation was observed between the absolute temperature difference (|ΔT|) and the proximity of each ROI. ROIs, regions of interest.

**Figure 6 bioengineering-10-00671-f006:**
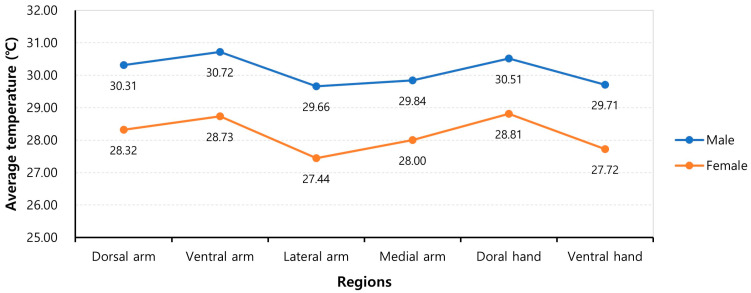
The average regional temperatures in men and women.

**Figure 7 bioengineering-10-00671-f007:**
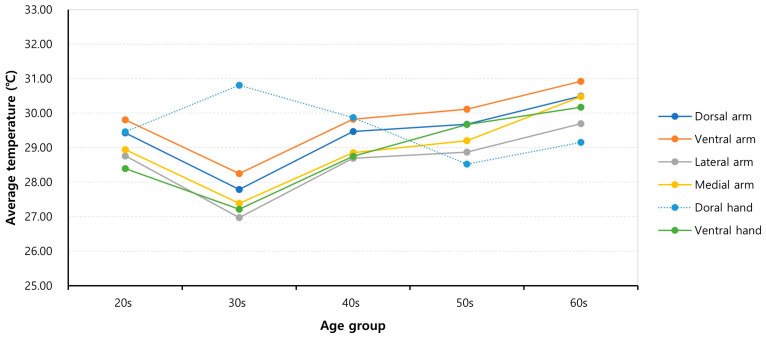
Relationship between the average regional temperatures and subject age.

**Table 1 bioengineering-10-00671-t001:** The average temperature of each region of interest in the dorsal arm region.

ROI	Mean (°C)	SD	Extended Uncertainty	Difference (ΔT, Right Side–Left Side)				
				Mean (°C)	SD	95% CI Lower	95% CI Upper	Extended Uncertainty
1_1	29.63	5.88	11.78	0.00				
1_2	29.74	5.93	11.88	−0.10	0.55	−0.22	0.01	1.35
2_1	29.96	5.93	11.88	0.00				
2_2	29.99	5.97	11.96	−0.03	0.54	−0.14	0.08	1.33
3_1	28.78	5.76	11.54	0.00				
3_2	28.82	5.77	11.56	−0.04	0.52	−0.15	0.06	1.30
4_1	29.18	5.80	11.62	0.00				
4_2	29.24	5.80	11.63	−0.06	0.49	−0.16	0.04	1.24
5_1	29.59	5.81	11.63	0.00				
5_2	29.62	5.82	11.67	−0.02	0.45	−0.12	0.07	1.17
6_1	29.26	5.81	11.64	0.00				
6_2	29.16	5.80	11.62	0.10	0.51	0.00	0.21	1.26
7_1	29.64	5.87	11.76	0.00				
7_2	29.72	5.94	11.90	−0.08	0.64	−0.21	0.06	1.48
8_1	30.24	5.90	11.82	0.00				
8_2	30.20	5.86	11.74	0.04	0.65	−0.09	0.17	1.49
9_1	29.66	5.86	11.74	0.00				
9_2	29.43	5.82	11.66	0.23	0.50	0.12	0.33	1.26
10_1	29.47	5.81	11.65	0.00				
10_2	29.21	5.77	11.56	0.25	0.48	0.15	0.35	1.22
11_1	29.22	5.73	11.48	0.00				
11_2	29.09	5.71	11.45	0.14	0.41	0.05	0.22	1.10
12_1	28.87	5.61	11.25	0.00				
12_2	28.77	5.61	11.24	0.11	0.52	0.00	0.21	1.28
13_1	28.97	5.62	11.26	0.00				
13_2	28.80	5.61	11.23	0.17	0.43	0.08	0.26	1.14
14_1	28.96	5.58	11.19	0.00				
14_2	28.81	5.56	11.14	0.15	0.43	0.06	0.23	1.13
15_1	28.56	5.42	10.87	0.00				
15_2	28.59	5.43	10.89	−0.03	0.42	−0.11	0.06	1.12
16_1	28.75	5.44	10.90	0.00				
16_2	28.69	5.43	10.89	0.06	0.44	−0.03	0.15	1.14
17_1	28.85	5.44	10.90	0.00				
17_2	28.72	5.43	10.89	0.13	0.44	0.03	0.22	1.15
18_1	28.88	5.49	11.01	0.00				
18_2	28.90	5.51	11.05	−0.01	0.47	−0.11	0.08	1.20
Mean	29.22	5.71	11.44	0.06	0.49	−0.05	0.16	1.24

CI, confidence interval; ROI, region of interest; SD, standard deviation.

**Table 2 bioengineering-10-00671-t002:** The average temperature of each region of interest in the ventral arm region.

ROI	Mean (°C)	SD	Extended Uncertainty	Difference (ΔT, Right Side–Left Side)				
				Mean (°C)	SD	95% CI Lower	95% CI Upper	Extended Uncertainty
1_1	29.81	5.90	11.82					
1_2	29.80	5.91	11.83	0.01	0.43	−0.08	0.10	1.13
2_1	29.88	5.91	11.84					
2_2	29.96	5.94	11.90	−0.08	0.44	−0.17	0.01	1.14
3_1	29.14	5.74	11.51					
3_2	29.16	5.73	11.48	−0.03	0.49	−0.13	0.07	1.23
4_1	29.60	5.77	11.57					
4_2	29.64	5.80	11.62	−0.04	0.46	−0.13	0.06	1.18
5_1	29.84	5.84	11.69					
5_2	29.86	5.88	11.78	−0.01	0.49	−0.11	0.09	1.24
6_1	29.76	5.83	11.67					
6_2	29.85	5.87	11.75	−0.08	0.54	−0.19	0.03	1.31
7_1	30.51	6.00	12.03					
7_2	30.47	5.96	11.94	0.04	0.51	−0.06	0.15	1.26
8_1	30.11	5.93	11.89					
8_2	29.92	5.84	11.71	0.19	0.54	0.07	0.30	1.32
9_1	29.12	5.70	11.42					
9_2	29.08	5.69	11.41	0.04	0.39	−0.04	0.12	1.07
10_1	29.46	5.70	11.43					
10_2	29.54	5.77	11.57	−0.07	0.42	−0.16	0.01	1.12
11_1	29.80	5.83	11.68					
11_2	29.73	5.83	11.67	0.08	0.41	−0.01	0.16	1.11
12_1	29.24	5.57	11.16					
12_2	29.42	5.65	11.33	−0.18	0.49	−0.29	−0.08	1.24
13_1	29.81	5.75	11.52					
13_2	29.84	5.77	11.57	−0.02	0.42	−0.11	0.07	1.13
14_1	29.69	5.76	11.55					
14_2	29.50	5.70	11.43	0.18	0.43	0.10	0.27	1.14
15_1	29.37	5.61	11.24					
15_2	29.53	5.65	11.32	−0.16	0.54	−0.27	−0.05	1.30
16_1	29.69	5.66	11.34					
16_2	29.59	5.63	11.28	0.10	0.52	−0.01	0.21	1.28
17_1	29.45	5.58	11.18					
17_2	29.27	5.49	11.00	0.18	0.59	0.06	0.30	1.39
18_1	29.45	5.51	11.04					
18_2	29.43	5.51	11.05	0.02	0.53	−0.09	0.13	1.30
19_1	29.42	5.50	11.02					
19_2	29.32	5.47	10.96	0.10	0.66	−0.03	0.24	1.51
Mean	29.63	5.74	11.51	0.01	0.49	−0.09	0.12	1.23

CI, confidence interval; ROI, region of interest; SD, standard deviation.

**Table 3 bioengineering-10-00671-t003:** The average temperature of each region of interest in the lateral arm region.

ROI	Mean (°C)	SD	Extended Uncertainty	Difference (ΔT, Right Side–Left Side)				
				Mean (°C)	SD	95% CI Lower	95% CI Upper	Extended Uncertainty
1_1	28.68	5.76	11.55					
1_2	28.88	5.88	11.79	−0.20	0.76	−0.36	−0.04	1.69
2_1	28.73	5.85	11.72					
2_2	28.62	5.86	11.75	0.12	0.56	0.00	0.23	1.34
3_1	28.63	5.84	11.70					
3_2	28.11	5.73	11.49	0.52	0.78	0.36	0.68	1.72
4_1	28.13	5.70	11.42					
4_2	28.19	5.74	11.51	−0.06	0.67	−0.19	0.08	1.52
5_1	28.28	5.73	11.48					
5_2	28.16	5.72	11.47	0.11	0.42	0.02	0.19	1.12
6_1	28.16	5.74	11.51					
6_2	27.88	5.67	11.37	0.28	0.56	0.16	0.40	1.35
7_1	28.64	5.81	11.63					
7_2	28.36	5.73	11.48	0.27	0.76	0.12	0.43	1.69
8_1	29.02	5.89	11.79					
8_2	28.73	5.83	11.68	0.29	0.63	0.16	0.42	1.46
9_1	28.46	5.79	11.60					
9_2	28.22	5.77	11.56	0.23	0.79	0.07	0.39	1.75
10_1	29.14	5.86	11.74					
10_2	28.71	5.79	11.61	0.43	0.67	0.29	0.57	1.54
11_1	29.53	5.90	11.82					
11_2	29.27	5.89	11.80	0.26	0.42	0.17	0.35	1.12
12_1	29.19	5.88	11.79					
12_2	29.11	5.90	11.82	0.07	0.59	−0.05	0.19	1.39
13_1	28.35	5.64	11.31					
13_2	28.12	5.61	11.25	0.22	0.71	0.07	0.37	1.59
14_1	28.76	5.71	11.44					
14_2	28.66	5.71	11.45	0.10	0.60	−0.03	0.22	1.41
15_1	28.47	5.65	11.32					
15_2	28.56	5.72	11.47	−0.10	0.70	−0.24	0.05	1.58
16_1	27.73	5.40	10.82					
16_2	27.59	5.39	10.80	0.14	0.65	0.01	0.28	1.49
17_1	28.01	5.44	10.90					
17_2	27.91	5.42	10.86	0.10	0.46	0.00	0.20	1.19
18_1	27.70	5.47	10.96					
18_2	27.63	5.44	10.91	0.06	0.69	−0.08	0.21	1.57
19_1	28.43	5.50	11.03					
19_2	28.37	5.50	11.03	0.05	0.45	−0.05	0.14	1.17
Mean	28.45	5.71	11.44	0.15	0.62	0.02	0.28	1.46

CI, confidence interval; ROI, region of interest; SD, standard deviation.

**Table 4 bioengineering-10-00671-t004:** The average temperature of each region of interest in the medial arm region.

ROI	Mean (°C)	SD	Extended Uncertainty	Difference (ΔT, Right Side–Left Side)				
				Mean (°C)	SD	95% CI Lower	95% CI Upper	Extended Uncertainty
1_1	28.60	5.48	10.98					
1_2	28.57	5.47	10.97	0.03	0.49	−0.07	0.13	1.24
2_1	28.68	5.43	10.88					
2_2	28.70	5.45	10.93	−0.01	0.45	−0.11	0.08	1.17
3_1	29.46	5.79	11.60					
3_2	29.36	5.76	11.54	0.10	0.45	0.01	0.20	1.17
4_1	28.85	5.57	11.16					
4_2	28.77	5.56	11.14	0.08	0.40	0.00	0.17	1.09
5_1	29.39	5.84	11.71					
5_2	29.31	5.80	11.63	0.08	0.43	−0.02	0.17	1.14
6_1	29.02	5.70	11.42					
6_2	28.91	5.66	11.34	0.11	0.42	0.02	0.20	1.12
7_1	28.00	5.23	10.50					
7_2	28.10	5.28	10.58	−0.10	0.69	−0.25	0.04	1.57
Mean	28.84	5.57	11.17	0.04	0.48	−0.06	0.14	1.21

CI, confidence interval; ROI, region of interest; SD, standard deviation.

**Table 5 bioengineering-10-00671-t005:** The average temperature of each region of interest in the dorsal hand region.

ROI	Mean (°C)	SD	Extended Uncertainty	Difference (ΔT, Right Side–Left Side)				
				Mean (°C)	SD	95% CI Lower	95% CI Upper	Extended Uncertainty
1_1	31.25	5.51	11.05					
1_2	31.11	5.53	11.08	0.13	0.44	0.04	0.22	1.15
2_1	31.27	5.53	11.08					
2_2	31.01	5.49	11.00	0.26	0.49	0.15	0.36	1.22
3_1	30.85	5.38	10.79					
3_2	30.73	5.37	10.76	0.11	0.59	−0.01	0.23	1.41
4_1	30.56	5.29	10.60					
4_2	30.60	5.31	10.65	−0.04	0.48	−0.14	0.06	1.21
5_1	30.76	5.32	10.67					
5_2	30.65	5.33	10.68	0.11	0.48	0.01	0.21	1.22
6_1	30.54	5.25	10.52					
6_2	30.36	5.21	10.46	0.18	0.52	0.07	0.29	1.28
7_1	29.93	5.06	10.15					
7_2	30.04	5.13	10.29	−0.11	0.45	−0.20	−0.01	1.17
8_1	30.34	5.13	10.29					
8_2	30.29	5.13	10.29	0.06	0.44	−0.03	0.15	1.16
9_1	30.37	5.14	10.32					
9_2	30.14	5.11	10.24	0.23	0.48	0.13	0.33	1.21
10_1	30.27	5.16	10.35					
10_2	30.31	5.22	10.46	−0.04	0.53	−0.15	0.07	1.29
11_1	30.47	5.08	10.20					
11_2	30.62	5.13	10.29	−0.16	0.50	−0.26	−0.05	1.25
12_1	30.64	5.52	11.07					
12_2	30.57	5.50	11.03	0.07	0.72	−0.07	0.22	1.62
13_1	30.05	5.13	10.29					
13_2	29.84	5.10	10.23	0.21	0.71	0.06	0.36	1.61
14_1	29.76	4.96	9.95					
14_2	29.69	4.97	9.98	0.06	0.64	−0.07	0.20	1.49
15_1	28.73	5.19	10.40					
15_2	28.77	5.23	10.48	−0.04	0.66	−0.18	0.10	1.51
16_1	28.33	5.05	10.13					
16_2	28.27	5.07	10.16	0.06	0.63	−0.07	0.19	1.47
17_1	28.15	4.91	9.84					
17_2	28.08	4.89	9.82	0.07	0.75	−0.09	0.23	1.67
18_1	29.26	4.96	9.94					
18_2	29.43	4.95	9.92	−0.17	0.57	−0.29	−0.06	1.36
19_1	28.89	4.96	9.95					
19_2	28.88	4.97	9.96	0.01	0.61	−0.11	0.14	1.43
20_1	28.77	5.13	10.29					
20_2	28.74	5.15	10.32	0.02	0.80	−0.14	0.18	1.76
21_1	29.28	4.96	9.95					
21_2	29.25	4.90	9.83	0.03	0.52	−0.08	0.14	1.28
22_1	28.99	5.07	10.17					
22_2	28.98	4.90	9.84	0.00	0.69	−0.14	0.14	1.56
23_1	29.11	5.12	10.28					
23_2	28.97	5.03	10.10	0.14	0.74	−0.02	0.29	1.65
24_1	29.29	4.87	9.77					
24_2	29.11	4.82	9.67	0.17	0.55	0.06	0.29	1.32
25_1	29.10	4.95	9.94					
25_2	29.02	4.86	9.74	0.08	0.64	−0.05	0.22	1.48
26_1	29.10	5.14	10.32					
26_2	28.88	5.05	10.13	0.21	0.79	0.05	0.38	1.75
Mean	29.74	5.14	10.30	0.06	0.59	−0.06	0.19	1.40

CI, confidence interval; ROI, region of interest; SD, standard deviation.

**Table 6 bioengineering-10-00671-t006:** The average temperature of each region of interest in the ventral hand region.

ROI	Mean (°C)	SD	Extended Uncertainty	Difference (ΔT, Right Side–Left Side)				
				Mean (°C)	SD	95% CI Lower	95% CI Upper	Extended Uncertainty
1_1	29.10	5.66	11.34					
1_2	29.09	5.67	11.36	0.01	0.40	−0.07	0.10	1.09
2_1	29.63	5.73	11.47					
2_2	29.71	5.77	11.57	−0.08	0.41	−0.17	0.00	1.11
3_1	29.73	5.80	11.62					
3_2	29.57	5.76	11.54	0.17	0.38	0.08	0.25	1.06
4_1	29.01	5.53	11.08					
4_2	29.13	5.59	11.20	−0.11	0.51	−0.22	−0.01	1.27
5_1	29.82	5.74	11.49					
5_2	29.85	5.76	11.55	−0.03	0.44	−0.13	0.06	1.16
6_1	29.53	5.70	11.43					
6_2	29.38	5.65	11.32	0.16	0.45	0.07	0.25	1.17
7_1	29.19	5.52	11.06					
7_2	29.33	5.60	11.23	−0.15	0.57	−0.26	−0.03	1.37
8_1	29.69	5.60	11.22					
8_2	29.62	5.59	11.21	0.07	0.57	−0.05	0.18	1.38
9_1	29.31	5.52	11.06					
9_2	29.12	5.50	11.02	0.19	0.68	0.05	0.33	1.55
10_1	28.16	5.13	10.29					
10_2	28.17	5.14	10.30	0.00	0.63	−0.14	0.13	1.47
11_1	28.70	5.33	10.68					
11_2	28.80	5.36	10.74	−0.10	0.83	−0.27	0.08	1.81
12_1	29.32	5.51	11.05					
12_2	29.30	5.57	11.16	0.02	0.73	−0.13	0.17	1.64
13_1	29.98	5.60	11.22					
13_2	29.88	5.60	11.22	0.10	0.63	−0.03	0.23	1.48
14_1	29.15	5.51	11.03					
14_2	29.09	5.47	10.96	0.06	0.86	−0.12	0.24	1.88
15_1	27.55	5.26	10.55					
15_2	27.49	5.32	10.67	0.06	0.87	−0.12	0.24	1.89
16_1	28.16	5.23	10.48					
16_2	28.05	5.31	10.65	0.11	0.84	−0.06	0.29	1.84
17_1	28.41	5.25	10.52					
17_2	28.37	5.27	10.57	0.04	0.69	−0.10	0.18	1.56
18_1	27.71	5.25	10.52					
18_2	27.52	5.21	10.44	0.19	0.82	0.02	0.36	1.80
19_1	28.16	5.31	10.64					
19_2	27.97	5.34	10.70	0.20	0.88	0.02	0.38	1.91
20_1	28.41	5.35	10.72					
20_2	28.30	5.36	10.74	0.11	0.74	−0.04	0.26	1.66
21_1	27.35	5.29	10.61					
21_2	27.26	5.23	10.49	0.09	0.84	−0.09	0.26	1.85
22_1	28.00	5.36	10.75					
22_2	27.88	5.25	10.54	0.12	0.85	−0.06	0.29	1.86
23_1	28.40	5.36	10.75					
23_2	28.20	5.30	10.63	0.20	0.76	0.04	0.36	1.70
24_1	27.25	5.16	10.35					
24_2	27.08	5.14	10.31	0.17	0.77	0.01	0.33	1.71
25_1	27.59	5.34	10.71					
25_2	27.41	5.33	10.69	0.19	0.79	0.02	0.35	1.75
26_1	27.84	5.48	10.98					
26_2	27.65	5.48	10.98	0.19	0.93	−0.01	0.38	2.01
Mean	28.62	5.44	10.91	0.07	0.69	−0.07	0.22	1.58

CI, confidence interval; ROI, region of interest; SD, standard deviation.

**Table 7 bioengineering-10-00671-t007:** Comparison of the average temperature in the same region of interest according to sex and age group.

Characteristics	Male (°C)	Female (°C)	Mean (°C)	Difference between Sexes (°C)	*p* Value
Dorsal arm region					<0.001 ^a^
20s (*n* = 183)	29.82 ± 5.37	28.99 ± 6.05	29.43 ± 5.69	0.83 (95% CI, 0.57–1.09)	<0.001 ^b^
30s (*n* = 213)	29.45 ± 5.83	26.06 ± 5.85	27.79 ± 5.84	3.39 (95% CI, 3.19–3.59)	<0.001 ^b^
40s (*n* = 228)	30.44 ± 5.55	28.58 ± 6.41	29.47 ± 6.00	1.86 (95% CI, 1.61–2.12)	<0.001 ^b^
50s (*n* = 177)	31.15 ± 5.04	28.82 ± 5.53	29.68 ± 5.35	2.33 (95% CI, 2.12–2.54)	<0.001 ^b^
60s (*n* = 104)	32.51 ± 4.59	29.59 ± 5.82	30.49 ± 5.44	2.92 (95% CI, 2.65–3.18)	<0.001 ^b^
Sum	30.31 ± 5.43	28.32 ± 5.94	29.22 ± 5.71	1.99 (95% CI, 1.76–2.22	<0.001 ^b^
Ventral arm region					<0.001 ^a^
20s (*n* = 183)	30.18 ± 5.41	29.40 ± 6.12	29.81 ± 5.74	0.78 (95% CI, 0.60–0.96)	<0.001 ^b^
30s (*n* = 213)	29.95 ± 5.93	26.51 ± 5.92	28.25 ± 5.93	3.44 (95% CI, 3.31–3.58)	<0.001 ^b^
40s (*n* = 228)	30.82 ± 5.59	28.90 ± 6.35	29.82 ± 5.99	1.92 (95% CI, 1.73–2.10)	<0.001 ^b^
50s (*n* = 177)	31.59 ± 5.05	29.25 ± 5.56	30.11 ± 5.37	2.34 (95% CI, 2.19–2.49)	<0.001 ^b^
60s (*n* = 104)	32.89 ± 4.55	30.05 ± 5.86	30.92 ± 5.46	2.84 (95% CI, 2.64–3.05)	<0.001 ^b^
Sum	30.72 ± 5.47	28.73 ± 5.97	29.63 ± 5.74	2.00 (95% CI, 1.84–2.16)	<0.001 ^b^
Lateral arm region					<0.001 ^a^
20s (*n* = 183)	29.18 ± 5.40	28.29 ± 6.16	28.76 ± 5.76	0.90 (95% CI, 0.66–1.14)	<0.001 ^b^
30s (*n* = 213)	28.76 ± 5.89	25.13 ± 5.85	26.97 ± 5.87	3.63 (95% CI, 3.43–3.83)	<0.001 ^b^
40s (*n* = 228)	29.85 ± 5.52	27.62 ± 6.31	28.69 ± 5.93	2.23 (95% CI, 1.99–2.48)	<0.001 ^b^
50s (*n* = 177)	30.57 ± 5.07	27.88 ± 5.52	28.87 ± 5.35	2.69 (95% CI, 2.48–2.90)	<0.001 ^b^
60s (*n* = 104)	31.67 ± 4.60	28.81 ± 5.73	29.69 ± 5.38	2.85 (95% CI, 2.58–3.13)	<0.001 ^b^
Sum	29.66 ± 5.45	27.44 ± 5.92	28.62 ± 5.67	2.22 (95% CI, 2.00–2.45)	<0.001 ^b^
Medial arm region					<0.001 ^a^
20s (*n* = 183)	29.25 ± 5.25	28.62 ± 5.97	28.94 ± 5.60	0.63 (95% CI, 0.24–1.01)	0.003 ^b^
30s (*n* = 213)	29.11 ± 5.72	25.80 ± 5.64	27.38 ± 5.68	3.15 (95% CI, 2.90–3.40)	<0.001 ^b^
40s (*n* = 228)	29.96 ± 5.44	28.21 ± 6.18	28.85 ± 5.91	1.74 (95% CI, 1.29–2.20)	<0.001 ^b^
50s (*n* = 177)	30.77 ± 4.98	28.50 ± 5.37	29.20 ± 5.25	2.27 (95% CI, 1.96–2.58)	<0.001 ^b^
60s (*n* = 104)	31.84 ± 4.56	29.34 ± 5.72	30.48 ± 5.19	2.51 (95% CI, 2.06–2.95)	<0.001 ^b^
Sum	29.84 ± 5.33	28.00 ± 5.78	28.84 ± 5.57	1.84 (95% CI, 1.49–2.20)	<0.001 ^b^
Dorsal hand region					<0.001 ^a^
20s (*n* = 183)	29.86 ± 5.31	29.04 ± 5.81	29.46 ± 5.56	0.82 (95% CI, 0.45–1.18)	<0.001 ^b^
30s (*n* = 213)	32.38 ± 3.83	29.38 ± 5.25	30.81 ± 4.57	3.00 (95% CI, 2.63–3.38)	<0.001 ^b^
40s (*n* = 228)	30.70 ± 4.92	28.93 ± 5.36	29.87 ± 5.13	1.77 (95% CI, 1.44–2.11)	<0.001 ^b^
50s (*n* = 177)	29.67 ± 5.33	27.34 ± 5.21	28.52 ± 5.27	2.33 (95% CI, 2.01–2.66)	<0.001 ^b^
60s (*n* = 104)	29.58 ± 5.51	28.76 ± 4.98	29.15 ± 5.23	0.83 (95% CI, 0.51–1.14)	<0.001 ^b^
Sum	30.51 ± 4.93	28.81 ± 5.38	29.74 ± 5.14	1.70 (95% CI, 1.36–2.05)	<0.001 ^b^
Ventral hand region					<0.001 ^a^
20s (*n* = 183)	29.12 ± 5.14	28.07 ± 5.79	28.39 ± 5.59	1.05 (95% CI, 0.70–1.40)	<0.001 ^b^
30s (*n* = 213)	29.28 ± 5.78	25.48 ± 5.43	27.21 ± 5.59	3.80 (95% CI, 3.49–4.10)	<0.001 ^b^
40s (*n* = 228)	29.52 ± 5.21	27.88 ± 5.80	28.75 ± 5.49	1.64 (95% CI, 1.27–2.02)	<0.001 ^b^
50s (*n* = 177)	30.86 ± 4.75	28.45 ± 5.49	29.67 ± 5.11	2.41 (95% CI, 2.12–2.70)	<0.001 ^b^
60s (*n* = 104)	31.30 ± 4.55	29.14 ± 5.67	30.17 ± 5.13	2.16 (95% CI, 1.74–2.59)	<0.001 ^b^
Sum	29.71 ± 5.22	27.72 ± 5.63	28.62 ± 5.44	2.00 (95% CI, 1.67–2.33)	<0.001 ^b^

^a^ analysis of variance, ^b^ paired *t*-test.

## Data Availability

The data presented in this study are available from the corresponding author on receipt of reasonable request and are not publicly available.

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
