# Peer review of "Reference Standards for Digital Infrared Thermography Measuring Surface Temperature of the Upper Limbs"

_bioengineering, 2023, doi:10.3390/bioengineering10060671_

Round 1

Reviewer 1 Report

The paper entitled "Reference standards for digital infrared thermography measuring surface temperature of the upper limbs" delas with the estabilishment of reference standards temperature distributions for the upper limbs. It seems to me very ambitious/risky to generalize the temperature distributions also if you considered a very big amount of subjects. I have expirience with medical thermal imaging and I can clearly assess that it is not possible to generalize in this field.  Body temperatures vary even in the same subject over the same day, especially for women, whose temoperature is highly dependent on the menstrual phase.

I think that you have to stress more this point in the limitation paragraph and also in the abstract.

The introduction is not so deepened. You may need to refer to other important works in the field (https://doi.org/10.3390/app11083614,  https://doi.org/10.1016/j.mvr.2015.08.008,https://doi.org/10.1186/s12891-023-06193-4,https://doi.org/10.1038/s41598-021-01381-5).

In the method section, a part deling with the methodological procedure is missing. For instance, the model of thermal camera used with its specifications, the distance from the subject, the time for acclimatation of the subjects, if any and the time necessary for the acquisition).

The results section can be improved with other graphics resuming data shown in the tables.

There are typos in lines 200 and 205--> RIO instead of ROI

The limitation paragraph as to be deeply revised, clearly stating that there are many confounders and citing related litterature.

The English languege is good.

Reviewer 2 Report

Comments:

In this manuscript, the authors proposed some reference standards for digital infrared thermography measuring surface temperature of the upper limbs. The authors recruited 905 healthy Korean adults and conducted thermography on six regions (dorsal arm, ventral arm, lateral arm, medial arm, dorsal hand, and ventral hand region) and analyzed the data based on the proximity of ROIs, sex, and age. Statistical results confirmed these data could be used as DITI reference standards to identify skin temperature abnormalities of the upper limbs. However, I have several concerns on this work detailed as below. Proper revision is suggested.

Here are some specific comments and suggestions:

1. How is sample size determined for the clinical trial? Why 905 volunteers? Is there any clinical scientific reason or proof? 

2. Why are these six regions (dorsal arm, ventral arm, lateral arm, medial arm, dorsal hand, and ventral hand region). chosen, and is there any scientific basis?

3. We know that the detection accuracy of digital infrared thermography measuring surface temperature is affected by the distance and human morphology of those volunteers. In this study, how did the authors ensure that the detection accuracy were not affected?

4. Whether the authors took into account the effect of the volunteers' height and weight on body temperature? I guess height and weight may have a big effect on body temperature.

5. It is suggested that the author further sort out the experimental data, and make a detailed description and summary.

 Moderate editing of English language

Round 2

Reviewer 1 Report

The authors have addressed every point I have raised. The work can be accepted in its current form in my opinion.

For me the quality of English language is good

Author Response

We appreciate reviewer’s recognition of the value of this article and our efforts in revising it. 

Reviewer 2 Report

The authors have responded to my comments on the work quite well, but I think the experimental method and result description can be further improved. 

Minor editing of English language required.

Author Response

We would like to express our gratitude for your thorough review and recommendations. We have incorporated the suggestions from the reviewer into our second revised manuscript, particularly in terms of describing the “3. Results” section in more detail. While we have limitations on providing a more extensive “2. Materials and Methods” section to avoid any potential plagiarism concerns related to our previous article on DITI of the lower extremities, we have made effort to provide a more comprehensive explanation in the "3. result" section. Please find the revised manuscript attached for your review. 

Round 3

Reviewer 2 Report

  • The relevant comments of reviewers have been reasonably interpreted and revised by authors.